# Development of Core Educational Content for Heart Failure Patients in Transition from Hospital to Home Care: A Delphi Study

**DOI:** 10.3390/ijerph19116550

**Published:** 2022-05-27

**Authors:** Seo-Jin Lee, Bo-Hwan Kim

**Affiliations:** 1Department of Nursing, Gachon University Graduate School, Incheon 21936, Korea; seojin8089@naver.com; 2College of Nursing, Gachon University, Incheon 21936, Korea

**Keywords:** heart failure, patient education, transitional care

## Abstract

Heart failure (HF) patients should be systematically educated before discharge on how to manage with standard written materials for patient self-management. However, because of the absence of readily available written materials to reinforce their learned knowledge, patients with HF feel inadequately informed in terms of the discharge information provided to them. This study aimed to develop core content to prepare patients with HF for transition from hospital to home care. The content was validated by expert panelists using Delphi methods. Nineteen draft items based on literature review were developed. We established a consensus on four core sections, including 47 categories and 128 subcategories through the Delphi survey: (1) understanding HF (five categories and 23 subcategories), (2) HF medication (19 categories and 45 subcategories), (3) HF management (20 categories and 47 subcategories), and (4) HF diary (three categories and 13 subcategories). Each section provided easy-to-understand educational contents using cartoon images and large or bold letters for older patients with HF. The developed core HF educational contents showed high consensus between the experts, along with clinical validity. The contents can be used as an educational booklet for both planning discharge education of patients with HF and for post-discharge management when transitioning from hospital to home. Based on this study, a booklet series for HF patients was first registered at the National Library of Korea. Future research should focus on delivering the core content to patients with HF in convenient and accessible format through various media.

## 1. Introduction

Several recent reports have emphasized that heart failure (HF) is a major global health problem, in that the overall number of people including older adults with HF has increased due to considerable increases in predisposing diseases or comorbidities [1,2,3]. Likewise, the incidence of patients with HF in South Korea was estimated to be 1.53% in 2013 and is expected to increase by 2.2-fold to 3.4% by 2040 [4]. Thus, over 1.7 million South Koreans are estimated to be affected by HF by 2040. Recently, in South Korea, the combined adoption of a more Western lifestyle and the rapidly aging population, as well as the increasing number of survivors of complex and serious cardiovascular illness due to recent advances in medical and surgical treatments, has resulted in a rapid increase in the prevalence of HF [4]. In Western countries, a higher prevalence of hospital readmissions has been observed to be mainly driven by worsening conditions of patients with HF during the post-discharge period. As a result, the 30-day readmission rate is approximately 18.2% and reaches 31.2% for the 90-day readmission rate [3,5,6]. These readmissions frequently occur if the patient fails to perform appropriate self-care including medication and dietary management, or if they miss the treatment period due to unrecognized symptom changes [7,8]. Repeated improvements and exacerbations in the disease progress of HF can reduce the patient’s quality of life and place physical and mental burdens on families or primary caregivers [9].

Although HF patients must respond to the diverse needs of individuals, the ultimate goal of HF patients is to acquire skills necessary to perform activities of daily living. In other words, to minimize repeated symptom exacerbations and prevent deterioration of the quality of life of patients and caregivers, it is necessary to improve the ability to recognize symptoms and manage HF [10,11]. To provide efficient and effective information to HF patients during the transition period from hospital to home, patients should be systematically educated before discharge on how to manage with standard written materials for patient self-management [11]. Although healthcare providers believed that they were fully prepared to provide information for patients with HF, because of the absence of readily available written materials to reinforce their learned knowledge, patients with HF feel inadequately informed in terms of the discharge information provided to them through various media [8,12]. As a result, it is difficult for patients with HF to continuously perform self-care due to insufficient information after conversion to home care [12].

The Delphi study is a group facilitation technique that aims to obtain consensus on the opinions of expert panelists through structured questionnaires [13]. It has been generally used in the field of health sciences education to encourage decision-making; the method uses a multistep technique of integrative processes intended to transform expert opinions into a finalized group consensus [14]. This method can help define the core educational content for HF patients and establish management preferences by anonymously reaching consensus among a panel of HF experts. 

This study aimed to develop core content to prepare patients with HF for the transition from hospital to home care using guidance obtained from the Delphi study. We would like to help patients maintain their pathophysiological health by developing an educational booklet on HF management and an HF diary that can be a guide to a healthy lifestyle for HF patients who transition from hospital to home care. In particular, we want to develop an educational booklet that can be easily understood and read by elderly HF patients.

## 2. Materials and Methods

### 2.1. Study Design

This is a Delphi study aimed at reaching consensus on the ideal core educational content of HF patients in the transition from hospital to home care from the perspective of South Korean experts.

### 2.2. Participants

We enlisted a homogeneous group of cardiac clinicians as the expert panel [15,16]. The criteria for selecting the expert panel were as follows: healthcare professionals with >10 years of experience and active participants in heart disease treatment, including HF. The eligibility criteria for the Delphi survey were as follows: (a) available to take part in three rounds [17,18] of the survey over a 2-month timeframe and (b) be fully fluent in using and having access to the internet and email using a computer or tablet. Those who refused to participate and who were not accustomed to using the internet were excluded.

Participants received an invitation email containing a letter introducing the Delphi survey and a written consent form regarding this study. The invitation email included detailed information about the purpose of the study, study design, survey procedures, deadlines, and average time required. In addition, we guaranteed the email panelists complete confidentiality regarding their involvement.

During the first Delphi round, we collected written informed consent and demographic data and explained to the participants that they had the freedom to choose to not complete the survey at any time in the process. They fully understood the purpose of the study and were willing participants. To compensate them for their time and efforts, all participants received a consulting fee after completing all rounds of the survey.

### 2.3. Research Procedure and Data Collection

#### 2.3.1. Development of Research Drafts

During the first step of the development of the accreditation standard for HF patients, we defined valid sources and methods. To develop core educational content, we included accreditation standards for HF from valid sources and methods. We used credible HF guidelines from the European Society of Cardiology [19,20], American Heart Association [21,22], and Korean Society of HF [23,24,25,26], and comprehensive education-related studies on the nature and different aspects of HF [7,10], and also investigated the ideas proposed by HF experts. It has been reported that if appropriate and evidence-based assessment standards are developed and applied to HF patients, they and their families may experience improved performance or management and high-quality services [27].

Based on this background and on a comprehensive review of the national and international literature, we devised four main sections: I. Understanding HF, II. HF medication, III. HF management, IV. HF diary. Drafts on 45 categories and 114 subcategories were prepared including the core educational contents for HF patients. First, the section Understanding HF included a general explanation of the definition, causes, symptoms, diagnosis, and treatment of HF, which is the basic content for understanding patients with HF. Second, in Section 2, “HF medication treatment,” the importance of HF medication treatment and type of medication (survival-enhancing drugs, symptomatic drugs, and other drugs) were classified and included. Third, in Section 3, “HF management,” taking important medications, monitoring and managing symptoms, maintaining daily life, and what caregivers need to know for self-management of HF patients were drafted. Fourth, in Section 4, “HF diary,” factors that should be measured for health management while practicing everyday activities were included. Since a majority of older patients were diagnosed with HF, we included cartoon pictures and photos for easy understanding by patients during clinical nursing practice.

#### 2.3.2. Delphi Survey

Round 1: To obtain approval of validated core educational items consisting of categories and subcategories from the above-mentioned sources, we asked the expert panel to confirm the items by open-ended questions. The expert panel received an email asking them to identify whether the list of the core educational content items was valid or not, and, if necessary, we asked the expert panel for additional content, or to correct any content that patients with HF should acquire before discharge. Items with acceptable or adaptable consensus and agreement were added to the list of validated items. The expert panel selected fundamental items and provided additional activities that patients with HF should perform during their daily living to recover their self-worth and also to assume responsibility for their health. As a result, the expert panel suggested 17 new items, and we added and constructed the core educational content items, as shown in Table 1.

Round 2: We surveyed the potential core HF educational content items identified in round 1 as the basis for round 2. This round was the first to rate the importance of items identified by the expert panel. We asked the expert panel to evaluate these items and to rate their importance for inclusion in the core educational content of HF using a five-point Likert scale (1 = totally disagree, 5 = strongly agree). We allowed the panel to explain their rating and asked them to suggest rephrasing of existing items or to propose new topics for consideration.

Expert panelists returned the email within 1 week, with reminders sent 5 days after the initial request. The results of the evaluation of this second round were described as medians and interquartile ranges for each item, and quantitative measures of consensus and agreement were used. We reached consensus that based the panel’s acceptance on whether an item should or should not be listed in the core educational content. We defined the validated core HF educational content items based on the accepted consensus items and agreement from the expert panel. Some items were suggested to be modified for HF patients and their families and were added after being qualitatively analyzed using the following procedure. After round 2, the finalized items and modified HF educational contents were subjected to round 3 for rating.

Round 3: In the third round of the Delphi surveys, we sent the expert panel the final items to be included as core educational contents that had been validated and compiled in round 2. After analyzing the results at each stage, the questionnaires were prepared for the next stage and were presented to the experts. The median and 25% and 75% quartile values derived from the round 2 Delphi survey results are presented for each item. This provided an opportunity for the panelists to enhance or revise their opinions by confirming the views of other experts [17,18]. The expert panel rated the items of the remaining and modified core HF educational contents using a five-point Likert scale. They responded within 1 week, with a reminder sent after 5 days. In the third round of the study, all 10 experts completed and returned the questionnaires.

### 2.4. Data Analysis

We used the SPSS/WIN 26.0 statistical package (version 26; IBM SPSS Statistics, Armonk, NY, USA) to analyze round 2 and 3 results via statistics as follows: frequencies, percentages, means, standard deviations (SDs), medians, and quartiles. The consensus of expert panelists was calculated using medians and interquartile ranges (1: interquartile range/median). We considered a “consensus” when each item was expressed by a median of 4.0 or greater or agreement of 75% or greater [17]. We also calculated content validity ratios (CVRs). The CVR is an item statistic that is useful in the rejection or retention of specific items, and the CVR varies between 1 and −1 [28]. We determined whether an item was necessary for operating a construct for a set of items using the CVR. Greater levels of content validity could exist when a larger number of expert panelists agreed that a particular item was essential. According to the criteria, the minimum CVR values of all items were set at 0.62 for 10 panel experts to meet the 5% level [28]. Stability in the Delphi survey method refers to the degree of agreement between the responses of the expert panel among repeated surveys. Therefore, stability was evaluated by the coefficient of variance (CV) calculated by dividing the SD by the mean [18].

### 2.5. Ethical Considerations

This study was approved by the Ethics Review Board of Gachon University (Approval No. 1044396-202102-HR-032-01). The authors explained the purpose of the study, voluntary nature of the participation in the interview, and absence of a penalty for not participating. All panelists provided their written informed consent electronically by completing a Consent Statement included in the introductory page of all surveys administered during the study. After round 3 was completed, a predetermined consulting fee was paid.

## 3. Results

### 3.1. Demographics of the Panel Experts

Because a homogeneous panel of 10–15 participants was deemed sufficient [15,16], we attempted to contact 12 experts via landline telephone using the intension sampling method. After explaining the purpose and method of the research, two experts refused to participate, and the resulting 10 experts participated in the study. The 10 experts who participated in this Delphi study completed all three surveys. None of them dropped out of the survey in the three rounds. The panel consisted of experts from health-related professions: two nursing professors, two cardiologists who are medical professors, three clinical nurses, one cardiac educational nurse, and two cardiac sonographers. They had all worked in the clinical and academic field of cardiology for over 10 years.

### 3.2. Results of Round 1

First, we drafted items that were to be provided to educate HF patients transitioning from hospital to home care using various evidence-based HF literature. We devised four sections, comprising 19 draft items including the following categories and subcategories: (1) understanding HF (five categories and 16 subcategories), (2) medical treatment of HF (19 categories and 45 subcategories), (3) management of HF (18 categories and 41 subcategories), and (4) HF diary (three categories and 13 subcategories). In Round 1 of the Delphi survey, we asked all panelists whether each draft item was includable or not or whether any item required correction. In addition, we attempted to obtain opinions and comments from the panelists using open-ended questions.

All panelists agreed with our developed draft items and provided comments which should be included in the transitional educational materials for HF patients and their caregivers. In the first section, “Understanding HF”, the expert panelists recommended the following: (1) atrial fibrillation should be included in the cause and risk factor category according to increased HF due to atrial fibrillation in the elderly, (2) dizziness and memory loss should be included in the signs and symptoms category, and (3) cardiac CT, cardiac magnetic resonance imaging (MRI), coronary angiography, exercise stress test, cardiac biopsy, and genetic testing should be added during screening and diagnosis. In the second section, “Medication treatment of HF”, no further comments were required. But some panelists suggested patient-tailored education for each medication because each patient would be prescribed different medications for the management of HF. In the third section, “Management of HF”, some panelists strongly underlined that it is very important not only to wash hands and get vaccinated but also to wear a mask and brush teeth to prevent flu and pneumonia. All panelists suggested that HF patients should assess activities of daily living and manage stress for self-management. The stress management should include how to overcome new stress and additional heart burden, physical changes under stress, how to reduce stress, muscle relaxation including suitable resting, and the importance of comfortable clothes. Some panelists asked to include that the primary caregiver should also carry out “checking and helping in the outpatient department (OPD) follow-up” for HF patients. In the fourth section, “HF diary”, all expert panelists agreed that a written daily entry, quick review and monitoring of HF symptoms, and a daily check list should be included for body weight, blood pressure and pulse rate, medication, fluid intake restriction, sodium intake monitoring, urine volume, exercise, no smoking, reduce or stop drinking, mood status, and monitoring of signs and symptoms typical of HF.

As indicated above, the expert panelists suggested 17 new items after round 1. According to the experts’ comments, we ultimate devised two categories and 15 subcategories. The two categories included “Activities of daily living” and “Stress management and overcoming stress”. The additional 15 subcategories were as follows: “Arterial fibrillation” in the cause and risk factors, “Dizziness and memory loss” in the sign and symptoms, “Cardiac computed tomography and MRI, coronary angiography, exercise stress test, cardiac biopsy, and genetic testing” in the screening and diagnosis, “Wearing mask and brushing teeth” in the importance of preventing flu and pneumonia, “Stress and heart burden, physical changes under stress, how to reduce stress, and muscle relaxation” in stress management, and “Checking and helping OPD follow-up” in the what a primary caregiver should do.

Taken together, we established a consensus of four core sections, 47 categories, and 128 subcategories. The sections were as follows: (1) understanding HF (five categories and 23 subcategories), (2) medication treatment of HF (19 categories and 45 subcategories), (3) management of HF (20 categories and 47 subcategories), and (4) HF diary (three categories and 13 subcategories).

### 3.3. Results of Rounds 2 and 3

After devising the 47 categories based on the results of round 1, a final agreement was reached regarding the validity of the categories for competence education of HF patients in rounds 2 and 3 (Table 2). For understanding HF, medication treatments of HF, management of HF, and HF diary, the CVR was 0.8 or higher and the content validity was verified for all categories. The lowest CVR was “sex and intimacy” in the management of HF content. In contrast, the others had higher CVR. However, the CVR scores were 0.8 and 1 for the categories; thus, these categories were not excluded. Subsequent to the second round, all categories reached a consensus. In round 2 of the Delphi survey, stability, which indicated the degree of agreement between the responses of the expert panel among repeated surveys, was measured by CV. The value of the lowest CV was 0.06 and the value of the highest CV was 0.14.

Compared with the results of round 2, in those of round 3,# the relevance of the items was relatively high and the expert panel’s responses appeared to have a relatively high stability CV from 0 to 0.14. All 47 categories analyzed had a minimum CVR of 0.8 or higher, and these results passed the minimum CVR of all items set at 0.62.

## 4. Discussion

The period of highest educational need for HF patients who want to know how to obtain a more balanced lifestyle is before discharge from hospital [29,30]. Therefore, to ensure that the patient can safely transition from hospital to home, transition nursing education is needed to advise patients on how to control their lifestyle from different points of view [31,32,33,34]. This study was conducted to obtain consensus opinions from practical and academic experts in HF to develop educational materials on healthy self-care management when patients transition from hospital to home care using a Delphi survey approach. The Delphi approach was suitable for surveying experts on this topic. By applying the Delphi survey, we obtained recommendations from multidisciplinary aspects to achieve a suitable quality of life for HF patients. Below, we will discuss our suggestions based on the consensus of the expert panel in this study.

Three HF self-care components were established: (1) maintenance in terms of adherence to treatment and maintaining healthy lifestyle behavior; (2) symptom perception such as response to symptoms when they occur; and (3) daily management in terms of good quality of life [35]. Therefore, in Section 1, “Understanding HF”, we considered “definition”, “cause and risk factors”, “signs and symptoms”, “screening and diagnoses”, and “treatment” as categories that are essential for HF patients [23,24,25]. Nurses should provide information to patients with HF to identify major risk factors for HF, as HF patients should be aware of the importance of active early treatment and self-management.

The ultimate goal of education when transitioning from hospital to home care is symptom perception and response to symptoms when they occur, and these play a key role in the self-care management of patients with HF to improve their health outcomes [35,36]. In these educational materials, the main symptoms of HF included edema (pulmonary edema and lower extremity edema) due to congestive HF and weight gain due to edema [23,24,25]. In addition, we included in the signs and symptoms related to decreased heart function a rapid heartbeat, fatigue, loss of appetite and indigestion, and dizziness and memory loss [23,24,25,26]. In addition, HF symptoms were explained with cartoon images so that heart dysfunction could be divided into left and right, and the severity of HF was also described according to the NYHA I to IV classes, which were easily understandable using images by anyone at a year/grade 5 to 6 reading level [37]. In particular, symptom monitoring for HF patients was divided into three zones: safety symptoms: green, warning symptoms: yellow, and emergency symptoms: red, and consisted of specific action phrases to encourage an action plan so that the patient could cope with his or her own condition [21,22]. This developed HF patient education material is superior to other educational materials in that it divided the patient’s symptoms into three stages (green, yellow, and red), which we used to educate patients to monitor their symptoms and perform an action plan according to the symptom stages.

There are many cases of repeated rehospitalization and discharge after acute and chronic HF that require sudden hospitalization in the early stages [38]. Rehospitalization means that worsening of symptoms has occurred, so active early clinical testing, treatment, and self-care of the patient are vital [39]. Various tests can be performed as active initial treatment, and since this method is crucial for understanding the basic conditions of the patient for diagnosis and treatment, the reason for performing various tests should be clearly explained to the patient [39]. Treatment of HF patients can be broadly divided into medication treatment, medical procedure and surgical treatment, and lifestyle adjustment [23,25,26]. In general, since most patients with HF are treated with two or three different medicines, an understanding of HF medications is required by patients [40] and patients must also be educated on the importance of medication adherence to ensure compliance [41]. Many HF patients are older aged adults and lack an understanding of the heart and its related organs (lungs, kidneys, and blood vessels), and the HF diagnosis-related instruments [37,40]. Therefore, to help them understand the clinical course of HF, we need easy-to-understand images to explain the mechanism of edema, a major symptom of HF, and thus, the heart structure with related organs was included in the educational material. In addition, as a textbook for older adults, large fonts are used on the whole, and bold fonts were used to emphasize important points [37].

In Section 2, “Medication treatments of HF”, we included all medications used to treat HF and tried to provide an easy explanation of the drugs’ mechanism of action in treating or controlling HF using cartoon images [40]. Even though so many different types of drugs are used in HF patients [23,25], it is inappropriate to provide patients with information on all drugs. Therefore, successful drug education of HF patients should be provided in a patient-specific manner [42,43]. Some drugs administered for treatment at hospitalization are given as drugs to be taken at home after discharge, so patients’ interest in the drugs and their ability to acquire information may increase. Therefore, patients with HF may benefit by support from bedside nurses who effectively assess patients’ discharge medications and educate them on each drug [44]. It is no exaggeration to say that self-management of HF patients begins with the ability to comply with taking medication [41]. Therefore, since drug administration in patient management is very important, we should plan to provide patient-tailored education materials for medication to each HF patient.

When returning to daily life after discharge, the most important aspect for managing HF patients is monitoring HF symptoms while taking medications regularly and achieving a balance in life according to the disease trajectory [11]. Therefore, here we developed Section 3 as “Management of HF”. The HF management section consists of how to assess the three so-called green, yellow, and red zones of HF, how to accurately use measuring devices such as a sphygmomanometer for blood pressure, how to monitor daily body weight to check for edema management such as weight gain or loss, and how to assess the degree of leg swelling [23,26]. To prevent HF, education should be provided on how to manage salt and water intake and output. In addition, many HF patients with diabetic mellitus should check blood sugar and concurrently control diabetes management [23,26].

In particular, the most difficult part of self-care in daily life is diet management [45]. Nevertheless, it is very important for HF patients to achieve a low salt/sodium intake [46], fluid restriction [47], and a low-fat diet [48], which will affect vascular health. In this educational booklet, information is organized so that individuals with HF can understand the basics of diet management and practice healthy eating habits. In other words, the following should be taught: the difference between salt and sodium, the principle of sodium intake’s aggravation of HF, sodium and water intake according to the severity of HF, low-salt diet, and avoiding processed food at home and when dining out, potassium intake and sodium excretion, basic information on low-fat diets and cholesterol, and overall dietary management for HF patients. This educational material suggests a dry diet method to help HF patients practice limiting sodium and water intake in their daily life. This is because, unlike Western food culture, Korean food culture involves a lot of broth, which can cause problems by increasing sodium and water intake.

Appropriate exercise should also be emphasized in maintaining a healthy lifestyle of HF patients [49]. HF patients often give up because they fear that exercise is too difficult due to major symptoms such as shortness of breath [50]. However, in patients with HF, exercise not only helps with symptom management but also plays an important role in maintaining a healthy lifestyle [49]. Exercise provides HF patients with ideal cardiac rehabilitation and should be tailored to the patient and performed under hospital prescription, but it is very difficult for most patients to return home and maintain exercise on their own [51]. Therefore, it can be effective form of education to help the HF patient consistently perform low- or mid-grade intensity exercise according to their symptom level [49]. This booklet also provides information to HF patients on the effects of smoking cessation, sobriety, work-life balance, maintaining a sex life, and the role of caregiver or family to determine how much help is needed after transition from hospital to home.

In addition, according to previous studies, depression and stress are very high in HF patients [52]. Especially during the COVID-19 pandemic, patients with HF are more likely to develop emotional stress, anxiety, and depression [53]. Because psychological and emotional assessment might be appropriate for self-care management, we included a checklist to determine the depression severity of the patient. We also included how to manage and overcome stress symptoms in the booklet, for example, participation in self-help groups, regular exercise and hobbies, and whole-body muscle relaxation training.

The four most important factors to prevent respiratory infections emphasized in this education booklet are hand washing, wearing a mask when going out, oral hygiene, and vaccination against influenza and pneumococcus. In particular, due to the COVID-19 pandemic, HF patients have a higher fatality rate due to the risk of respiratory infection than those with conditions [53]. Therefore, washing hands and wearing a mask are very important [54]. In particular, these respiratory infections lower lung function, aggravate HF, and put more strain on the heart, so education has been focused on preventing influenza and pneumonia [55]. Considering the current COVID-19 pandemic, this educational material can be an important part of a self-care guide for HF patients.

In Section 4, “HF diary”, the first part is structured to include a patient’s pledge, and the second part is a quick review of HF warning signs and emergency symptoms. In this last section, a checklist evaluating daily weight, signs of edema, blood pressure, and pulse, taking medicine, daily water intake, eating salty food notes, urine volume, exercise, smoking and drinking cessation, mood status, and signs and symptoms is provided. Through this diary, patients will be able to check their daily trajectory of health conditions of HF for 100 days.

Based on this study, a booklet series for HF patients was first registered at the National Library of Korea. An educational booklet series, under the title of *A Guide to Healthy Living for HF Patients: Understanding and Management of HF* was published, combining Sections 1 and 2. Section 2 was published under the title of *A Guide to Healthy Living for HF Patients: Medication Treatment of HF*, which contains all the information on medication types used in patients with HF. When transition nursing intervention is provided, tailored medication information will be provided to the patient, and contents of the medication booklet will be revised and provided individually to the patient, because the medication information should be customized for each patient and configured differently for each patient. Section 4 was also published as *A Guide to Healthy Living for HF Patients: My HF Diary* for patients with HF to check their health condition daily. Use of the diary might help HF patients balance their lifestyle, even if they are on the trajectory of the disease process.

### Limitations

We recruited expert panelists using targeted convenient extraction for sampling, which means that it was impossible to guarantee the inclusion of all potential experts in the HF field. We attempted to attract a range of experts to ensure a varied and in-depth analysis of each topic, encompassing both clinical and educational aspects. Thus, the recruitment was targeted at both physicians and registered nurses. Registered nurses and physicians came from various healthcare units (e.g., outpatient education units, echocardiographic units, critical care units, and general wards) from five hospitals and two nursing colleges. Moreover, we tried to decrease the impact of possible local institutional policies on the goals and content of education both in terms of their organization and geography. Only experts in HF care were recruited as panelists, and the majority of the nurses and physicians who participated in the study specialized in HF care, which increased the reliability of their comments.

The Delphi survey was performed as an online survey, and all of the expert panelists participated in it separately. Unlike a face-to-face Delphi survey approach, which is performed sitting at a round table, an online Delphi ensures anonymity for panelists, which encourages honesty and reduces the risk of dominant or high-profile participants controlling the discussion [56]. As a result, the panelists could not discuss or interact for consensus online, and this may have lowered the validity of this study, although concurrently all the panelists participated from the first to third round, which might have increased the validity of this study. We did not evaluate the stability between the second and third rounds because consensus between the experts was already high in the second round and it improved in the third round. Although this study was conducted in South Korea, which means that the results cannot be generalized directly to other countries, we attempted to include HF self-care education contents considering the standard guidelines of different countries. To improve the completeness of this study, a qualitative study in the form of an interview on the evaluation of preceding HF education materials, or a Delphi study on the constituted educational booklet including both heart failure patients and caregivers, is necessary. In future research on the development of HF educational materials, collection of opinions from receivers of the education is warranted.

## 5. Conclusions

Four competence sections with 47 categories and 128 subcategories were developed for core HF educational contents by a Delphi panel of heart experts. The consensus between the expert panelists was high, which suggests that the educational contents were clinically valid. This Delphi result could be used as a content for both discharge education and post-discharge management when transitioning from hospital to home care. Based on this study, a booklet series for HF patients was first registered at the National Library of Korea. Future research should focus on delivering the developed content to patients with HF as convenient and accessible education through various media.

## Figures and Tables

**Table 1 ijerph-19-06550-t001:** Final content of heart failure discharge education after the first round.

Section	Categories and Subcategories	
I. Understanding HF	1. Definition	
		(1) Anatomy and physiology of the heart, (2) Mechanism of disease
	2. Cause and risk factor	
		(1) Coronary diseases, (2) Cardiomyopathy, (3) Valvular diseases (4) Hypertension, (5) Atrial fibrillation *
	3. Signs and symptoms	
		(1) Main symptoms: *(1) dyspnea, (2) increasing body weight, (3) edema, (4) fatigue, (5) cough with sputum, (6) tachycardia, (7) anorexia and indigestion, (8) dizziness and memory loss **, (2) Symptoms of right and left HF, (3) NYHA functional classification (I to IV)
	4. Screening and diagnosis	
		(1) History taking and physical examination, (2) Electrocardiography (3) Chest X-ray, (4) Blood test (including BNP or NT-proBNP), (5) Echocardiography, (6) Cardiac CT *, (7) Cardiac MRI *, (8) Coronary angiography *, (9) Exercise stress test *, (10) Cardiac biopsy *, (11) Genetic testing *
	5. Treatment	
		(1) Medications, (2) Medical and surgical procedures: *(1) valve replacement, (2) percutaneous coronary intervention, (3) coronary artery bypass graft, (4) cardiac resynchronization therapy, (5) intracardiac defibrillator, (6) ventricular assist device, (7) heart transplantation*
II. Medication treatments of HF	1. Importance of medication treatments	
	2. Medications for HF in major organs	
		(1) Heart and coronary artery, (2) Kidney, (3) Blood vessels
	3. Type of HF medications	
		(1) Medication for increasing survival rate: *(1) angiotensin-converting enzyme inhibitor, (2) angiotensin receptor blocker, (3) beta blocker, (4) angiotensin receptor-neprilysin inhibitor, (5) mineralocorticoid antagonist, (6) selective If channel blocker*, (2) Medication for reliving signs and symptoms: *(1) diuretics, (2) cardiotonic, (3) Others: (1) anticoagulant, (2) calcium channel blocker, (3) combination tablets, (4) vasodilator*
	4. Angiotensin-converting enzyme inhibitor	
		(1) Mechanism of drug action, (2) Name (generic and trade)
	5. Angiotensin receptor blocker	
		(1) Mechanism of drug action, (2) Name (generic and trade)
	6. Beta Blocker	
		(1) Mechanism of drug action, (2) Name (generic and trade)
	7. Angiotensin receptor neprilysin inhibitor	
		(1) Mechanism of drug action, (2) Name (generic and trade)
	8. Mineralocorticoid antagonist	
		(1) Mechanism of drug action, (2) Name (generic and trade)
	9. Selective if channel blocker	
		(1) Mechanism of drug action, (2) Name (generic and trade)
	10. Diuretics	
		(1) Mechanism of drug action, (2) Name (generic and trade), (3) Adverse action and medication interactions related with potassium ion
	11. Cardiotonic	
		(1) Mechanism of drug action, (2) Name (generic and trade), (3) Adverse action and therapeutic digoxin level
	12. Anticoagulant	
		(1) Mechanism of drug action, (2) Name (generic and trade), (3) Adverse action, therapeutic level, and medication interactions related with Vitamin K
	13. Calcium channel blocker	
		(1) Mechanism of drug action, (2) Name (generic and trade)
	14. Combination tablets	
		(1) Mechanism of drug action, (2) Name (generic and trade)
	15. Vasodilator	
		(1) Mechanism of drug action, (2) Name (generic and trade), (3) Nitroglycerin and use
	16. How to take medications	
		(1) Every day right on time and right methods, (2) Do not stop taking medicines by yourself
	17. Identifying patients’ medications	
		(1) Medication related with HF, (2) Other medications
	18. Monitoring side effects of medication treatments	
		(1) Dizziness, (2) Hyperkalemia, (3) Dry cough related to angiotensin-converting enzyme, (4) Dehydration related with diuretics, (5) Bradycardia related with beta blocker
	19. Medication interactions	
		(1) Other medications: OTC drug, (2) Foods, health functional foods, and health supplements
III. Management of HF	1. Monitoring signs and symptoms	
		(1) Stable status, (2) Caution status, (3) Emergency status
	2. Monitoring blood pressure	
		(1) Importance of checking BP, (2) How to measure BP, (3) Precaution when measuring BP
	3. Monitoring body weight for body fluid	
		(1) Importance of checking BW, (2) Ideal dry BW
	4. Monitoring edema	
		(1) How to assess edema, (2) How to control edema
	5. Monitoring urine volume	
		(1) Checking urine volume
	6. Monitoring blood glucose	
	7. Restricting fluid intake	
		(1) How much water fluid is restricted?
	8. Restricting sodium intake	
		(1) How much sodium is restricted? (2) Relation between salt and sodium, (3) Low-salt diet at home and on eating out, (4) Sodium ranking in food
	9. Restricting fat and cholesterol	
		(1) Relation between cardiovascular health and fat/cholesterol, (2) Type of cholesterol and normal range, (3) How to control fat and cholesterol in foods
	10. HF patient-tailored diet control	
		(1) Selecting foods by each food group, (2) Question and answer when difficulty in eating meal, (3) How to read nutrient labeling in processed foods: sodium
	11. Exercise for HF	
		(1) Importance of exercise, (2) Recommended exercise, (3) Cautions for exercising, (4) Stretching before exercise
	12. No smoking	
		(1) Health effects by duration of smoking cessation
	13. Reduc or stop drinking	
		(1) Alcohol effects on the heart, (2) Caloric ranking in alcohol beverage
	14. Importance of preventing flu and pneumonia	
		(1) Hand washing, (2) Wearing mask *, (3) Brushing teeth *, (4) Vaccination
	15. Work-life balance in HF	
	16. Sex and intimacy after HF	
	17. Checking activities of daily living (K-IADL) *	
	18. Expression the emotions	
		(1) Importance of expressing the emotions, (2) Checking the emotions (PHQ-9)
	19. Stress management and overcome *	
		(1) Stress and heart burden *, (2) Physical changes under stress *, (3) How to reduce stress *, (4) Muscle relaxation *, (5) Suitable rest and comfortable clothes, (6) Resting
	20. What a primary caregiver should do	
		(1) Checking if patient takes medications, (2) Observing for patient change signs and symptoms, (3) Assisting in low-salt diet control, (4) Helping with physical activity and exercise, (5) Emotional support, (6) Checking and helping with OPD follow-up *
IV. HF diary	1. Written oath	
	2. Review of monitoring HF symptoms	
		(1) Caution condition status, (2) Emergency condition status
	3. Daily check	
		(1) Body weight, (2) Blood pressure and pulse rate, (3) Taking medicines, (4) Restricting fluid intake, (5) Monitoring sodium intake, (6) Urine volume, (7) Exercise, (8) No smoking, (9) Reduce or stop drinking, (10) Mood status, (11) Monitoring signs and symptoms

* Newly added items in round 1. BNP, b-type natriuretic peptide; BP, blood pressure; BW, body weight; CT, computed tomography; HF, heart failure; K-IADL, Korea instrumental activities of daily living; MRI, magnetic resonance imaging; NT-proBNP, n-terminal prohormone of brain natriureptic peptide; NYHA, New York heart association; OTC, over-the-counter; PHQ-9, patient health questionnaire-9.

**Table 2 ijerph-19-06550-t002:** Results of the second and third rounds of the Delphi survey (N = 10).

Contents	Categories	Second Round			Third Round		
		M ± SD	CVR	CV	M ± SD	CVR	CV
I. Understanding HF	1. Definition	4.50 ± 0.527	1	0.12	4.70 ± 0.483	1	0.10
	2. Cause and risk factors	4.50 ± 0.527	1	0.12	4.70 ± 0.483	1	0.10
	3. Signs and symptoms	4.50 ± 0.527	1	0.12	4.80 ± 0.422	1	0.09
	4. Screening and diagnosis	4.40 ± 0.516	1	0.12	4.70 ± 0.483	1	0.10
	5. Treatment	4.40 ± 0.516	1	0.12	4.70 ± 0.483	1	0.10
II. Medication treatments of HF	1. Importance of medication treatments	4.90 ± 0.316	1	0.06	4.90 ± 0.316	1	0.06
	2. Medications for HF in major organ	4.70 ± 0.483	1	0.10	4.90 ± 0.316	1	0.06
	3. Type of HF medications	4.70 ± 0.483	1	0.10	4.90 ± 0.316	1	0.06
	4. Angiotensin-converting enzyme inhibitor	4.80 ± 0.422	1	0.09	4.90 ± 0.316	1	0.06
	5. Angiotensin receptor blocker	4.80 ± 0.422	1	0.09	5.00 ± 0.00	1	0.00
	6. Beta blocker	4.80 ± 0.422	1	0.09	4.90 ± 0.316	1	0.06
	7. Angiotensin receptor neprilysin inhibitor	4.80 ± 0.422	1	0.09	4.90 ± 0.316	1	0.06
	8. Mineralocorticoid antagonist	4.80 ± 0.422	1	0.09	5.00 ± 0.00	1	0.00
	9. Selective sinus node I(f) channel inhibitor	4.80 ± 0.422	1	0.09	4.90 ± 0.316	1	0.06
	10. Diuretics	4.80 ± 0.422	1	0.09	4.90 ± 0.316	1	0.06
	11. Cardiotonic	4.80 ± 0.422	1	0.09	4.90 ± 0.316	1	0.06
	12. Anticoagulant	4.60 ± 0.516	1	0.11	4.90 ± 0.316	1	0.06
	13. Calcium channel blocker	4.80 ± 0.422	1	0.09	4.90 ± 0.316	1	0.06
	14. Combination tablets	4.80 ± 0.422	1	0.09	4.90 ± 0.316	1	0.06
	15. Vasodilator	4.80 ± 0.422	1	0.09	5.00 ± 0.00	1	0.00
	16. How to take medications	4.80 ± 0.422	1	0.09	4.90 ± 0.316	1	0.06
	17. Identifying patients’ medications	4.90 ± 0.316	1	0.06	4.90 ± 0.316	1	0.06
	18. Monitoring side effects of medication treatments	4.80 ± 0.422	1	0.09	4.90 ± 0.316	1	0.06
	19. Medication interactions	4.60 ± 0.516	1	0.11	4.80 ± 0.422	1	0.09
III. Management of HF	1. Monitoring signs and symptoms	4.90 ± 0.316	1	0.06	4.90 ± 0.316	1	0.06
	2. Monitoring and checking blood pressure	4.90 ± 0.316	1	0.06	4.90 ± 0.316	1	0.06
	3. Monitoring and checking body weight for body fluid	4.80 ± 0.422	1	0.09	4.90 ± 0.316	1	0.06
	4. Monitoring and assessment of edema	4.90 ± 0.316	1	0.06	4.90 ± 0.316	1	0.06
	5. Monitoring and checking urine volume	4.80 ± 0.422	1	0.09	4.90 ± 0.316	1	0.06
	6. Monitoring and checking blood glucose	4.70 ± 0.483	1	0.10	4.70 ± 0.483	1	0.10
	7. Restricting fluid intake	4.90 ± 0.316	1	0.06	4.90 ± 0.316	1	0.06
	8. Restricting sodium intake: at home, at eating out	4.80 ± 0.422	1	0.09	4.90 ± 0.316	1	0.06
	9. Restricting fat and cholesterol	4.80 ± 0.422	1	0.09	4.90 ± 0.316	1	0.06
	10. HF patient-tailored diet control	4.70 ± 0.483	1	0.10	4.80 ± 0.422	1	0.09
	11. Exercise for HF	4.70 ± 0.483	1	0.10	4.90 ± 0.316	1	0.06
	12. No smoking	4.80 ± 0.422	1	0.09	5.00 ± 0.00	1	0.00
	13. Reduce or stop drinking	4.80 ± 0.422	1	0.09	5.00 ± 0.00	1	0.00
	14. Importance of preventing flu and pneumonia	4.90 ± 0.316	1	0.06	4.90 ± 0.316	1	0.06
	15. Working	4.90 ± 0.316	1	0.06	4.90 ± 0.316	1	0.06
	16. Sex and intimacy	4.70 ± 0.675	0.8	0.14	4.70 ± 0.675	0.8	0.14
	17. Checking activities of daily living (K-IADL)	4.90 ± 0.316	1	0.06	4.90 ± 0.316	1	0.06
	18. Expression emotions	4.80 ± 0.422	1	0.09	4.80 ± 0.422	1	0.09
	19. Stress management	4.80 ± 0.422	1	0.09	5.00 ± 0.00	1	0.00
	20. What a primary caregiver should do	4.80 ± 0.422	1	0.09	4.90 ± 0.316	1	0.06
IV. HF diary	1. Written oath	4.70 ± 0.483	1	0.10	4.70 ± 0.483	1	0.10
	2. Review of monitoring HF symptoms	4.80 ± 0.422	1	0.09	5.00 ± 0.00	1	0.00
	3. Daily check for 100 days	4.80 ± 0.422	1	0.09	5.00 ± 0.00	1	0.00

CV, coefficient of variance; CVRs, content validity ratios; HF, heart failure; K-IADL, Korea instrumental activities of daily living.

## Data Availability

Not applicable.

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
