# Peer review of "Development of Core Educational Content for Heart Failure Patients in Transition from Hospital to Home Care: A Delphi Study"

_ijerph, 2022, doi:10.3390/ijerph19116550_

Round 1

Reviewer 1 Report

The design and implementation of the study is excellent. It shows a high level of methodological mastery. The application of the technique is flawless. Only the conclusions could be explained a little more fully, not so schematically. The rest of the article is simply perfect. Congratulations

Author Response

Response to Reviewers’ Comments

We wish to thank you for your thoughtful comments and valuable feedback on our manuscript titled Development of Core Educational Content for Heart Failure Patients in Transition from Hospital to Home Care: A Delphi Study”

We have revised the manuscript according to the Reviewers’ suggestions. We have rewritten and rephrased sections to improve clarity and added further information to explain details regarding some vague points.

For your convenience, we have used red font for the revisions. Please find the following revisions according to Reviewers’ comments.

Best regards

----------------------------------------------------------------------------------------

Reviewer #1’s comments:

The design and implementation of the study is excellent. It shows a high level of methodological mastery. The application of the technique is flawless. Only the conclusions could be explained a little more fully, not so schematically. The rest of the article is simply perfect. Congratulations.

Response : Thank you for your reviewing the manuscript and your congratulations.

Reviewer 2 Report

Thank you for giving me a chance to review the manuscript, “Development of Core Educational Content for Heart Failure Patients in Transition from Hospital to Home Care: A Delphi Study” Please see my comments below.

Title: delete the “.” At the end of title

Abstract:

  • Should follow the journal guideline with headings, including introduction, methods, results, conclusion
  • Any pilot study was conducted to examine the effect of the content or how did you know the effectiveness of the content

Introduction

I wonder why this study did not include patients with HF as they are the population to receive the educational content, especially when the authors argue that inadequate information to support the self-care of this patient population

Materials and methods

  • The educational content was for the HF patients. I concern about the applicability of the educational content.
  • Only healthcare professionals who were although experienced, may not representative enough to understand the needs of the HF patients.
  • Although Delphi study will include relevant parties to discuss and design  and then modify that repeat few rounds to consolidate the product. However, the panel if only include cardiac clinicals but not other healthcare professionals and even HF patients, how did the authors know the needs of this HF population. What areas do they need to have more information about self-care.
  • The educational materials to enhance HF patients can be specific for HF patients in a country. Identifying the specific needs and difficulties in self care of the HF patient is more important.
  • The final content is fit for the clinicians but not really for the HF patients. A HF patient who does not have valvular disease will not care how this disease impact his HF condition.

Overall

Although the study area to develop an educational material to enhance self-care of HF patient is important, the approach or design was not appropriate to address the study aim. The quality of this study is very low and not valid enough.

Author Response

Response to Reviewers’ Comments

We wish to thank you for your thoughtful comments and valuable feedback on our manuscript titled Development of Core Educational Content for Heart Failure Patients in Transition from Hospital to Home Care: A Delphi Study”

We have revised the manuscript according to the Reviewers’ suggestions. We have rewritten and rephrased sections to improve clarity and added further information to explain details regarding some vague points.

For your convenience, we have used red font for the revisions. Please find the following revisions according to Reviewers’ comments.

Best regards

----------------------------------------------------------------------------------------

Reviewer #2’s comments:

Thank you for giving me a chance to review the manuscript, “Development of Core Educational Content for Heart Failure Patients in Transition from Hospital to Home Care: A Delphi Study” Please see my comments below.

Comment 1. Title: delete the “.” At the end of title

Response : Thank you for your comment. We deleted “.” at the end of the title.

  •  Development of Core Educational Content for Heart Failure Patients in Transition from Hospital to Home Care: A Delphi Study

Comment 2. Abstract:

Should follow the journal guideline with headings, including introduction, methods, results, conclusion

Response : Thank you for your comment. We have followed the guidelines of IJERPH and accordingly added the appropriate headings.

Comment 3. Any pilot study was conducted to examine the effect of the content or how did you know the effectiveness of the content

Response : Thank you for your comment. Currently, we are conducting a study on patients with heart failure to confirm the effect of this study.

Comment 4.

Introduction

I wonder why this study did not include patients with HF as they are the population to receive the educational content, especially when the authors argue that inadequate information to support the self-care of this patient population.

Materials and methods

The educational content was for the HF patients. I concern about the applicability of the educational content.

Only healthcare professionals who were although experienced, may not representative enough to understand the needs of the HF patients.

Although Delphi study will include relevant parties to discuss and design and then modify that repeat few rounds to consolidate the product. However, the panel if only include cardiac clinicals but not other healthcare professionals and even HF patients, how did the authors know the needs of this HF population. What areas do they need to have more information about self-care.

The educational materials to enhance HF patients can be specific for HF patients in a country. Identifying the specific needs and difficulties in self-care of the HF patient is more important.

The final content is fit for the clinicians but not really for the HF patients. A HF patient who does not have valvular disease will not care how this disease impact his HF condition.

Overall

Although the study area to develop an educational material to enhance self-care of HF patient is important, the approach or design was not appropriate to address the study aim. The quality of this study is very low and not valid enough.

Response : Thank you for your comment. This research team reviewed all previous studies since a long time on heart failure patients in our country and the educational brochures provided to them. As a result, there were educational materials such as simple leaflets provided to patients, but there were no educational materials to help patients read and practice. Therefore, this study focused on the development of an educational booklet that is easy to comprehend and follow with pictures for ease of understanding of patients and practice of heart failure self-care. We added the following to the limitations of this study as follows:

  • “To improve the completeness of this study, a qualitative study in the form of an interview on the evaluation of preceding HF education materials or a Delphi study on the constituted educational booklet including both heart failure patients and caregivers is necessary. In future research on the development of HF educational materials, collection of opinions from receivers of the education is warranted.”

Reviewer 3 Report

This work is meaningful to implement for HF patients in clinic. I hope you to revise this manuscript as my comments for enhancing it. 

In introduction part on page 2, Aims of this paper were confused. Please rewrite them. For example,

line 64, 'this study ultimately aims to~' comparing line 69

line 71~73, there are 3 aims including to select, to identify, and to extract. So, give each number for each aim or to unify them.  

In line 84 of participants part, you have to show the references for the number of panel  

Especially, in line 87, you described process with 3 rounds, so you have to show the references also. 

In line 140, the table 1 is not fancy but so long. Please modify it with compact, attractive, readable, and clear. I recommend to merge some lines for making just one page. 

Author Response

Response to Reviewers’ Comments

We wish to thank you for your thoughtful comments and valuable feedback on our manuscript titled Development of Core Educational Content for Heart Failure Patients in Transition from Hospital to Home Care: A Delphi Study”

We have revised the manuscript according to the Reviewers’ suggestions. We have rewritten and rephrased sections to improve clarity and added further information to explain details regarding some vague points.

For your convenience, we have used red font for the revisions. Please find the following revisions according to Reviewers’ comments.

Best regards

----------------------------------------------------------------------------------------

Reviewer #3’s comments:

This work is meaningful to implement for HF patients in clinic. I hope you to revise this manuscript as my comments for enhancing it.

Comment 1. In introduction part on page 2, Aims of this paper were confused. Please rewrite them. For example, line 64, 'this study ultimately aims to~' comparing line 69, line 71~73, there are 3 aims including to select, to identify, and to extract. So, give each number for each aim or to unify them.

Response : Thank you for your valuable comment. We have revised the text to clarify the aim of this paper and for ease of reader comprehension.

  • “This study aimed to develop core content to prepare patients with HF for the transition from hospital to home care using guidance obtained from the Delphi study. We would like to help patients maintain their pathophysiological health by developing an educational booklet on HF management and an HF diary that can be a guide to a healthy lifestyle for HF patients who transition from hospital to home-care. In particular, we want to develop an educational booklet that can be easily understood and read by elderly HF patients.”

Comment 2. In line 84 of participants part, you have to show the references for the number of panel

Response : Thank you for your valuable comment. We have inserted two references.

  • “We enlisted a homogeneous group of cardiac clinicians as the expert panel [15, 16].”

Comment 3. Especially, in line 87, you described process with 3 rounds, so you have to show the references also.

Response : Thank you for your valuable comment. We have inserted two references.

  • “The eligibility criteria for the Delphi survey were as follows: (a) available to take part in three rounds [17, 18] of the survey over a 2-month timeframe and (b)”

Comment 4. In line 140, the table 1 is not fancy but so long. Please modify it with compact, attractive, readable, and clear. I recommend to merge some lines for making just one page.

Response : Thank you for your valuable comment. We have modified Table 1 to make it more compact, attractive, readable, and clear by merging some lines. Please see Table 1 in the manuscript.

Round 2

Reviewer 2 Report

Thanks for the revised manuscript. This version is acceptable. Hope the following qualitative study to understand the evaluation of effectiveness of the education can help the HF patients better.